# A Study on the Super Resolution Combining Spatial Attention and Channel Attention

Dongwoo Lee [1], Kyeongseok Jang [1], Soo Young Cho [2], Seunghyun Lee [3] and Kwangchul Son [2],*

[1] Department of Plasma Bio Display, Kwangwoon University, 20 Gwangun-ro, Nowon-gu, Seoul 01897, Republic of Korea
[2] Department of Information Contents, Kwangwoon University, 20 Gwangun-ro, Nowon-gu, Seoul 01897, Republic of Korea
[3] Ingenium College, Kwangwoon University, 20 Gwangun-ro, Nowon-gu, Seoul 01897, Republic of Korea
* Correspondence: kcson@kw.ac.kr; Tel.: +82-2-940-8100

**Abstract:** Existing CNN-based super resolution methods have low emphasis on high-frequency features, resulting in poor performance for contours and textures. To solve this problem, this paper proposes single image super resolution using an attention mechanism that emphasizes high-frequency features and a feature extraction process with different depths. In order to emphasize the high-frequency features of the channel and space, it is composed of CSBlock that combines channel attention and spatial attention. Attention block using 10 CSBlocks was used for high-frequency feature extraction. In order to extract various features with different degrees of feature emphasis from insufficient low-resolution features, features were extracted from structures connected with different numbers of attention blocks. The extracted features were expanded through sub-pixel convolution to create super resolution images, and learning was performed through $L_1$ loss. Compared to the existing deep learning method, it showed improved results in several high-frequency features such as small object outlines and line patterns. In PSNR and SSIM, it showed about 11% to 26% improvement over the existing Bicubic interpolation and about 1 to 2% improvement over VDSR and EDSR.

**Keywords:** super resolution; channel attention; spatial attention; parallel structure

## 1. Introduction

Recently, demand for high-resolution (HR) images is increasing due to the development of high-resolution displays such as TVs, smartphones, and monitors. However, there are limitations in obtaining a large number of high-resolution images due to limitations of high-resolution cameras, storage devices, and transmission and reception equipment. To solve this problem, a single image super resolution (SISR) method was used to restore an HR image from a single low-resolution (LR) image. However, since SISR can structurally have one LR image for several HR images, it is an ill-posed problem to which there is no single correct answer. Due to this ill-posed problem, SISR configures HR image and LR image as a pair and proceeds with restoration. As an ill-posed problem, SISR has been studied for decades [1,2].

SISR is largely classified into two technologies. The first is an interpolation method that estimates empty pixels generated in the process of expanding an image by referring to neighboring pixel values [3]. The second method is to generate new pixel values through deep learning-based CNN [4]. For interpolation, there are representative methods such as nearest neighbor interpolation and bicubic interpolation [5]. Nearest neighbor interpolation is a method of estimating the value of an empty pixel with one nearest pixel value. It can be executed quickly through a small number of calculations, but has a problem of generating block-shaped artifacts. In order to solve this problem, bicubic interpolation is a method of estimating the value of an empty pixel by using the distance and three-dimensional function of the information of 16 neighboring pixels based on the empty

pixel. This minimizes block-shaped artifacts and shows a more natural result than nearest neighbor interpolation. However, the same calculation is performed on high-frequency components such as outlines and textures, resulting in blurry texture expressions. On the other hand, the deep learning-based method can accurately generate empty pixel values through more complex and diverse operations than the existing interpolation-based method. Deep learning-based methods generate new pixel values through CNN, and include SRCNN [6], VDSR [7], and EDSR [8]. SRCNN is a model that proposes end-to-end learning based on deep learning. It uses bicubic interpolation-extended images as input and consists of three convolution layers. On the other hand, VDSR is a deep network model and consists of 20 convolution layers deeper than the three layers of SRCNN. Effective learning is possible through residual learning that adds input images and output images. EDSR is a 69-layer network structure that is deeper than VDSR, and efficient learning was conducted using residual blocks [9]. Existing deep learning-based methods create a network without considering high-frequency components such as outlines and textures, which makes it difficult to create clear images. Recently, an attention-based super resolution method using such high-frequency feature emphasis has been proposed [10,11].

In this paper, SISR using attention is proposed to solve the high-frequency region and feature extraction problems that were not considered in the existing CNN-based super resolution method. In order to emphasize high-frequency features, feature enhancement for high-frequency features was carried out with CSBlock, which combines channel attention and spatial attention, and various features were utilized through feature extraction processes of various depths. In addition, the deep network structure helps improve performance, but there is a gradient vanishing/exploding problem, which is a difficult learning problem. The feature map extracted in this way was extended using sub-pixel convolution, and an HR image was created through residual block and convolution. In order to learn this structure, it was composed of a $48 \times 48$ size LR patch and an HR patch, and learning was conducted using $L_1$ Loss. Through this structure, natural HR images were created by emphasizing high-frequency features that greatly affect image quality and utilizing various features.

## 2. Related Work

### 2.1. ResNet

ResNet is a convolution neural network model using skip connection to effectively learn a neural network deeper than the existing 19 layer VGG19 neural network [12]. Figure 1 is a ResNet structure consisting of 152 convolution layers.

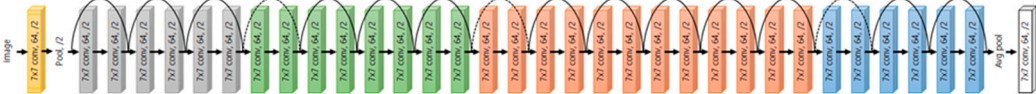

**Figure 1.** ResNet Architecture.

In general, as the number of layers increases in a neural network, a higher level of features can be extracted. However, when the neural network increases beyond a certain layer, the gradient vanishing problem occurs, and learning is not performed properly [13]. ResNet uses residual block using skip connection to solve this problem.

$$F(x) = H(ReLU(H(x)), \tag{1}$$

$$R(x) = H(ReLU(H(x)) + x, \tag{2}$$

(a) is the existing neural network and (b) is the residual block structure proposed by Resnet in Figure 2. Equation (1) is the existing neural network formula, and Equation (2) is the residual block formula of ResNet. In Equations (1) and (2), $x$ is the input feature map, *ReLU* is the Rectified Linear Unit, and $H$ is the convolution. Existing neural networks train the neural network to output from input $x$ feature maps [14]. However, in ResNet, unlike existing neural networks, the neural network is trained so that the sum of the

output and the input comes from the input $x$ feature map. ResNet is a structure that learns $R(x) - x = H(ReLU(H(x)))$ to learn the weight of $H(ReLU(H(x)))$, and there is a detour path from input to output. This path is called skip connection, and it is called identity mapping because the conversion input state is maintained through this path. In ResNet, it is called residual block because it learns the difference between input and output with this skip connection. If an input similar to the output comes in, the convolution operation of the residual block learns a value close to the zero vector, so even a small change in the input can be sensitively detected. In addition, if the target value has already been extracted in the previous step as a deep layer, it can be used as an output by skipping several convolutions with a skip connection structure through the convolution operation. Therefore, one neural network has the effect of using multiple neural networks with different numbers of layers. ResNet can learn neural networks with many layers and increase accuracy by using residual blocks with such a skip connection.

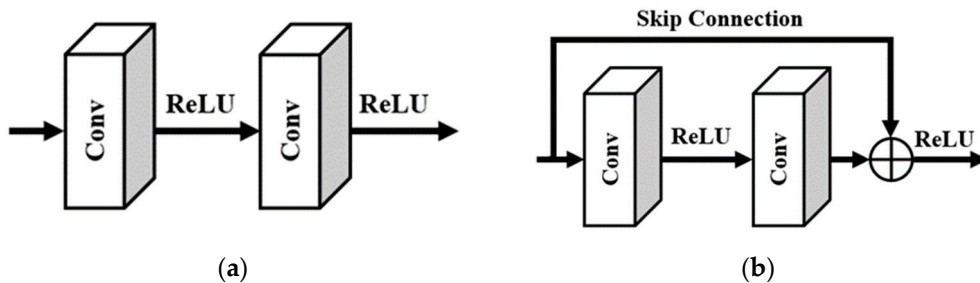

**Figure 2.** Network Model: (**a**) Existing Neural Networks; (**b**) ResNet's Residual Block.

### 2.2. Attention Mechanism

Seq2Seq is a model used in natural language processing, such as machine translation, that uses an encoder–decoder architecture to transform one sequence into another [15]. However, during the process of compressing the encoder–decoder structure of the seq2seq model into a fixed-size vector, loss of information can occur and the gradient vanishing problem can occur when the input is long. To address these issues, attention can be used effectively to train the model.

Recently, attention has been applied as self-attention in CNN as well as seq2seq. In the field of computer vision such as object detection and semantic segmentation, objects to be detected and segmented are distributed in a specific area, not in the entire image [16–19]. Features are extracted by emphasizing a specific region and ignoring the remaining unnecessary regions. These features extract location and semantic information-oriented features and utilize the semantic and location features of objects to improve network performance and show good performance even in noisy inputs. Attention used in CNN includes convolutional block attention module (CBAM) [20]. Figure 3 is the architecture of CBAM.

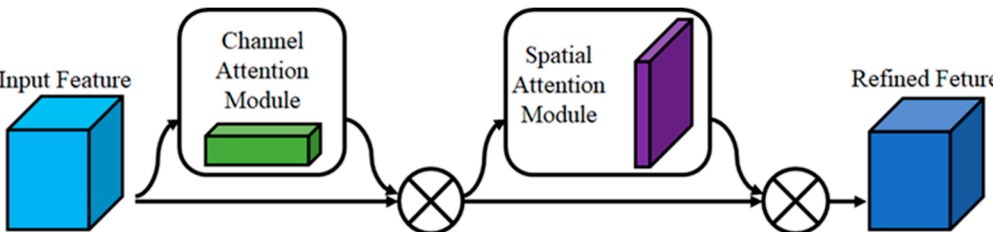

**Figure 3.** CBAM Architecture.

Figure 4 is the channel attention module of CBAM. Channel attention creates an attention map by utilizing the relationship between channels of features. Channel attention compresses the space through *MaxPool* and *AvgPool* to efficiently perform calculations. The channel feature association between each feature map is calculated through a multi-layer

perceptron (MLP) of the compressed features. Afterwards, an attention map is created with the sum of the two outputs and the sigmoid between 0 and 1.

$$M_C = \sigma(MLP(AvgPool(F)) + MLP(MaxPool(F))), \tag{3}$$

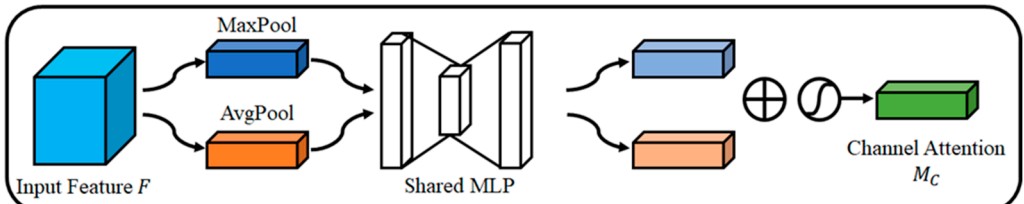

**Figure 4.** Channel Attention Module.

Equation (3) is the channel attention module of CBAM. $\sigma$ is the sigmoid activation function, *MLP* is the multi-layer perceptron, *AvgPool* is average pooling, *MaxPool* is max pooling, and *F* is the input feature map.

Figure 5 is the structure of spatial attention. Spatial attention creates an attention map by utilizing the relationship between spaces. Spatial attention creates a one-channel feature map through convolution. Through this, unlike channel attention, relation between pixels is calculated. Afterwards, an attention map is created with a sigmoid value between 0 and 1.

$$M_S = \sigma\left(f^{7 \times 7}([AvgPool(F); MaxPool(F)])\right), \tag{4}$$

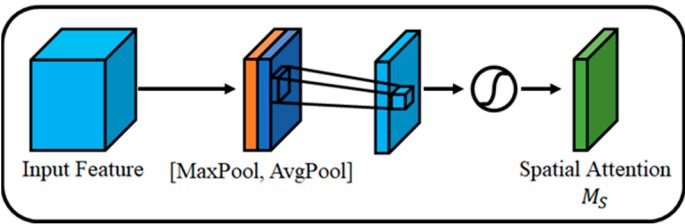

**Figure 5.** Spatial Attention Module.

Equation (4) is the spatial attention module of CBAM. $f^{7 \times 7}$ is a $7 \times 7$ convolution.

*2.3. Sub-Pixel Convolution*

Sub-pixel convolution [21] rearranges the feature map to extend the feature map like deconvolution [22,23]. Super resolution is a kind of reasoning problem that restores information lost in HR images by using the redundancy of LR images. In order to utilize this redundancy, a large number of convolutions are used or features are extracted using convolutions with various filters. This causes the problem that the network becomes complicated or a large number of parameters are used. To solve this problem, sub-pixel convolution rearranges several feature maps to create extended feature maps. This structure increases the redundancy of LR patches rather than one deconvolution. Recently, a super resolution method using sub-pixel convolution has been applied, and a method of reducing the amount of calculation and parameters by using an LR image as an input image is being studied.

Figure 6 is the process of sup-pixel convolution; a is the input feature map, b is the feature map resulting from the convolution, and c is the feature map rearranged from the convolution feature map. For rearraying in sub-pixel convolution, the number of channels equal to the square of the expansion coefficient is required.

$$f^{ps}(x) = PS(W * x + b), \tag{5}$$

Equation (5) is an equation for the sub-pixel convolution process. *PS*() is the rearrangement of the LR feature map, *W* is the weight, *x* is the input feature map, and b is the

bias. $PS(T)_{x,y,z}$ is rearranged for expansion during the sub-pixel convolution process and is expressed in the following formula.

$$PS(T)_{x,y,c} = T_{\lfloor x/r \rfloor \lfloor y/r \rfloor, C \cdot r \cdot mod(y,r) + C \cdot mod(x,r) + c}, \tag{6}$$

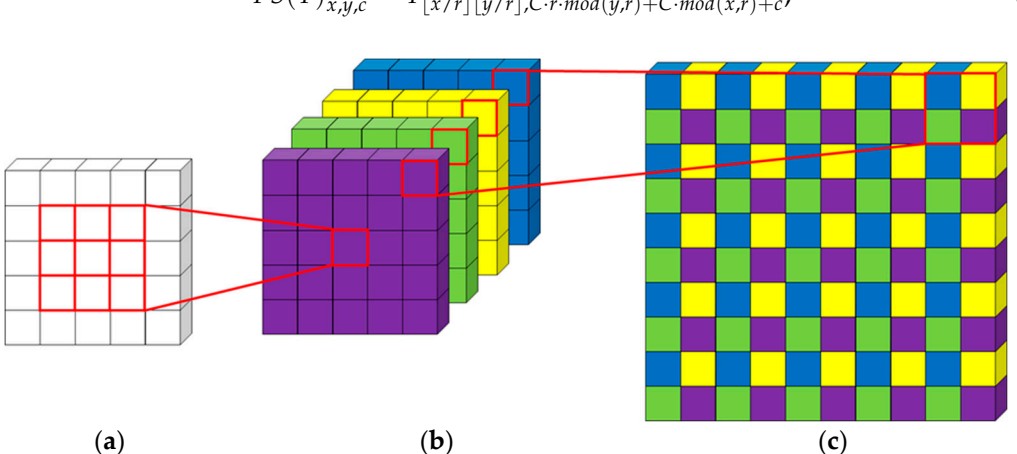

**(a)**        **(b)**        **(c)**

**Figure 6.** Spatial Attention Module: (**a**) Input Feature Map; (**b**) the feature map resulting from the convolution; (**c**) feature map rearranged from the convolution feature map.

In Equation (6), $T$ is the tensor of the feature map, $x$, $y$, $c$ are the $x$, $y$, and $z$ axes of the input $T$, and $C$ is the channel of the extended feature map. To expand a feature map of size $x * y * c$ to $n \cdot x * n \cdot y * c$, the number of channels of the output feature map needs to be $c \cdot n^2$. where $n$ is the expansion factor. To this end, the number of channels of the feature map must be increased through convolution, and the size of the feature map is expanded through rearrangement. Through this process, in this paper, it was used to expand feature maps to secure redundancy, and high-quality, HR images were created by resolving checkerboard artifact.

## 3. Proposed Method

This paper proposes a super resolution method using CSBlock and various feature extractions for high-frequency enhancement in images. In general, image quality has a great influence on high-frequency features, and for this purpose, the high-frequency features are emphasized by applying a CSBlock structure that combines channel attention and spatial attention. To enable effective training of deep networks, an attention block combining CSBlock and skip connection was applied. The attention block was used to emphasize features using the structure of Shallow Feature Extraction and Deep Feature Extraction with attention block, and feature utilization was performed through connections. In this way, both shallow features including shape information and deep features with emphasized high-frequency components were extracted from the insufficient information of low resolution. Then, super resolution was performed by expanding the feature maps through the upsampling process of various feature maps with high-frequency features emphasized. Figure 7 is the structure of the proposed super resolution.

### 3.1. CSBlock

Existing convolution operation is a method for measuring the similarity between the center pixel value and the kernel pattern, and it outputs a higher value as it is similar to the kernel pattern. However, there is a problem in that patterns of pixels exceeding the size of the kernel cannot be extracted by multiplying the surrounding pixel values by the kernel weight. To solve this problem, pixel features can be extracted by increasing the size of the kernel or by overlapping multiple convolutions. However, many parameters and calculations are required, and the uncertainty of location information means it does not know where the feature that is extracted appears. This is further manifested in a high-frequency feature, which is a part in which a difference between a pixel value at a specific

location and a neighboring pixel value is large. This problem is solved using CSBlock, which emphasizes high-frequency features.

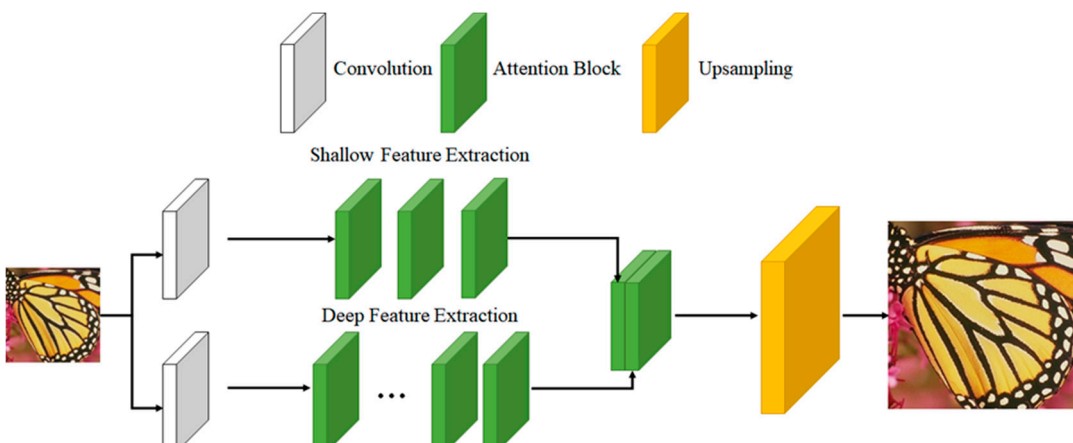

**Figure 7.** Proposed Method Architecture.

Figure 8 shows the structure of CSBlock that combines channel attention and spatial attention. In the first step, the input feature map was reduced through global average pooling (GAP) [24]. The relation and importance between channels was calculated using the fully connected layer [25] for the reduced feature map in the second step, and channel attention was performed through multiplication with the input feature map. In the third step, a feature map was created by calculating the relation and importance between pixels through one-channel convolution. Finally, spatial attention was performed by multiplying the feature map, which calculated the relation and importance between pixels, with the channel attention-progressed feature map.

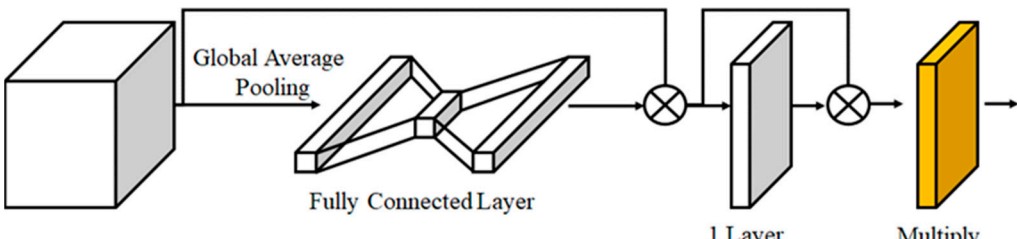

**Figure 8.** Channel Spatial Block.

Figure 9 is the channel attention structure of the proposed method. Unlike the existing CBAM, the MAX Pool part was removed and only the average pool was applied. Through this structure, only the strong high frequency was emphasized, and it was calculated as a value between 0 and 1 by sigmoid operation. With this structure, efficient channel attention was possible with the amount of computation.

$$H_{GP}(x) = \frac{1}{H \times W} \sum_{i=1}^{H} \sum_{j=1}^{W} x(i,j), \tag{7}$$

Equation (7) is GAP, which is a process of extracting feature values for each channel to proceed with channel attention. $H$ and $W$ mean the height and width of the feature map, respectively. For $H_{GP}(*)$, one channel of the feature map is averaged with one value and expressed as a representative feature value for each channel. In addition, a large number of operations and parameters are required to calculate feature maps without using GAP. GAP was used to represent a representative feature map including semantic information for each channel while reducing the number of parameters and calculations.

$$s(x) = f(W_d \times \delta(W_{d/4} \times H_{GP}(x))), \tag{8}$$

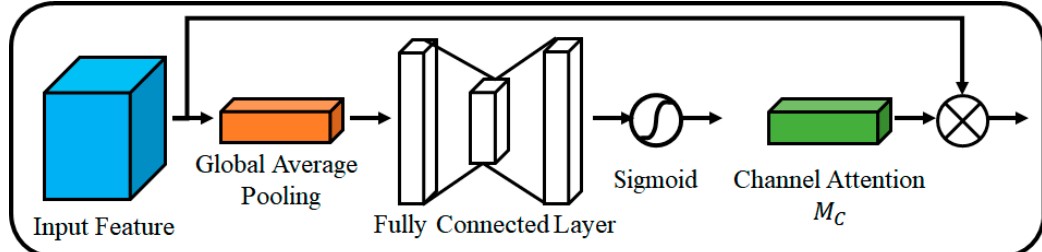

**Figure 9.** Channel Attention Structure.

Equation (8) calculates relation between channels and channel importance in the GAP obtained from $H_{GAP}(*)$ in Equation (7). f and δ refer to the sigmoid and ReLU activation functions, respectively, and $W_d$ and $W_{d/4}$ are the weights of the fully connected layer with $d$ and $d/4$ channels in the output feature map, respectively. Relation and importance between channels were calculated through two fully connected layers. Afterwards, the sigmoid activation function was calculated as an attention value between 0 and 1. The amount of computation and the number of parameters are reduced through the fully connected layer, which reduces the number of channels in the middle of the channel of the feature map.

$$CA(x) = x \times s, \tag{9}$$

Equation (9) generates an emphasized feature map by multiplying the channel attention feature map calculated through Equation (8) with the input feature map. $x$ is the input feature map, and $s$ is the channel attention feature map obtained from Equation (11). The high-frequency features included in the feature map were emphasized and feature map channels containing unnecessary features were suppressed. The channel attention map is output between 0 and 1 through the sigmoid activation function, necessary channels are passed through multiplication with the input feature map, and unnecessary channels are suppressed through a number close to 0.

Figure 10 is the spatial attention structure of the proposed method. Unlike the existing CBAM, the output of convolution was composed of one channel without pooling process. This reduces spatial information loss that may occur during the pooling process and emphasizes spatial features.

$$SA(x) = H_1(x) \times x, \tag{10}$$

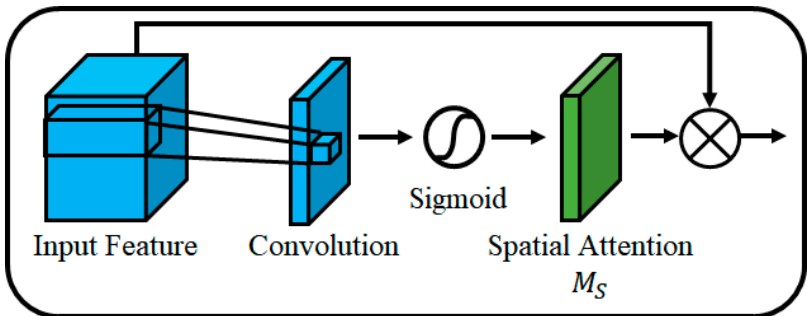

**Figure 10.** Spatial Attention Structure.

Equation (10) generates an accentuated feature map by multiplying the spatial attention feature map with the input feature map. $H_1(*)$ is a one-channel convolution operation, and $x$ is the input feature map. Convolution is a process of calculating surrounding pixels into a kernel, and can use spatial information. $H_1(*)$ calculates the relation between pixels in the feature map. Through channel attention, feature maps with high frequencies are emphasized, and features that lack high frequencies or are unnecessary are suppressed. In the highlighted feature map, the part with a large difference in neighboring pixel values was effectively calculated through spatial attention.

$$CSBlock(x) = SA(CA(x)) \times residual\ scale, \tag{11}$$

Equation (11) is the formula of the final *CSBlock* to which channel attention and spatial attention are applied. *SA*(*) is the spatial attention process of Equation (10), and *CA*(*) is the channel attention process of Equation (9). Stable learning was conducted using the residual scale. This structure effectively solves the problem of inability to learn or performance degradation that occurs in the process of learning a deep network by limiting the value of the gradient.

### 3.2. Attention Block

Figure 11 is Attention of the proposed method. The high-frequency region in an image is the outline and texture information of an object and has a great influence on the visual quality of an image. This means that high-frequency region restoration is important in the super resolution process, and for this purpose, high-frequency features are emphasized using CSBlock that uses attention mechanism. It is possible to emphasize strong high-frequency features by connecting CSBlocks in series, and a skip connection structure is applied to solve the gradient vanishing problem that can occur in the deepened network due to serial connection. Through this structure, learning was conducted without emphasizing high-frequency features and without a gradient vanishing problem.

$$AB(x) = H(CSBlock_{10}(CSBlock_9(\cdots CSBlock_1(x)\cdots)) + x), \tag{12}$$

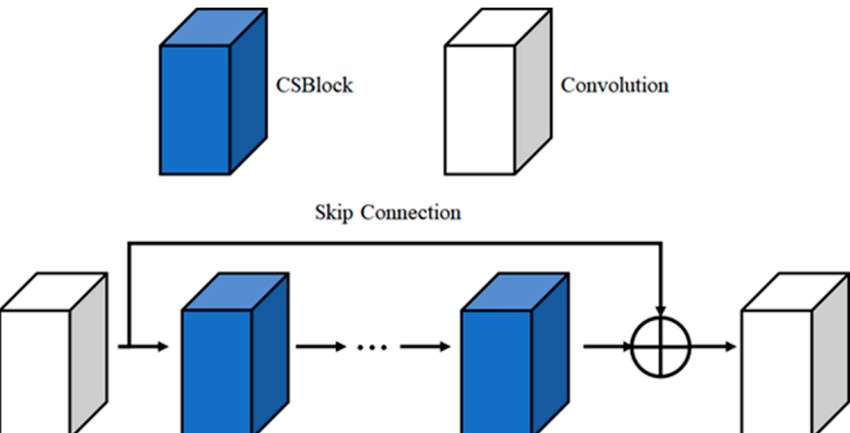

**Figure 11.** Attention Block.

Equation (12) is an attention block formula for emphasizing super resolution. $AB(*)$ means attention block, $H(*)$ means convolution operation, $CSBlock_n(*)$ is the nth CSBlock, and $x$ is the input feature map. The network used 10 CSBlocks and was composed of a skip connection structure and convolution. This structure helps network learning by preventing and supplementing gradient vanishing.

### 3.3. Feature Extraction

LR images have less information due to the difference in resolution compared to HR images, and it is important to extract and utilize various features. In order to extract various features, the attention blocks were extracted at different depths. The following Equation (13) is used to extract various features.

$$F_n(x) = AB_n(AB_{n-1}(\cdots AB_1(x))), \tag{13}$$

$F_n(x)$ feature extraction, $AB_n(x)$, is the nth attention block and $x$ is the input feature map. Attention block is a structure that suppresses unnecessary information and emphasizes important information to focus on a specific part in the input feature map. If the structure of the attention block is shallow, the shape can be distinguished, but there is

the problem that it is greatly affected by surrounding noise, and if the structure is deep, high frequencies such as the contour stone or texture of the object are emphasized and may cause loss of positional information. In order to extract various features, two depth structures were used in the attention block. After making attention blocks with three and 10 structures, feature extraction was performed.

$$F_s(x) = AB_3(AB_2(AB_1(x))),$$ (14)

$$F_d(x) = AB_{10}(AB_9(\cdots AB_1(x))),$$ (15)

In Equation (14), $F_s(x)$ uses three attention blocks to distinguish the shape, and $F_d(x)$ in Equation (15) is a formula using 10 attention blocks to extract various features of the object. After that, in order to effectively utilize the features extracted from various depths, each extracted feature map was connected through concatenation. With this concatenation, various features can be used and shared, and fast learning is possible.

### 3.4. Upsampling

Deconvolution was used to extend feature maps in the existing super resolution process. However, the deconvolution process causes checkerboard artifact problems due to redundant calculations. To solve this effectively, the feature map was extended using sub-pixel convolution. Afterwards, RGB super resolution images were created through residual blocks and three-channel convolution. Figure 12 is an extension process image for each feature map extension.

$$F_{up} = Up([F_s(H(I_{lr}), F_D(H(I_{lr}))]),$$ (16)

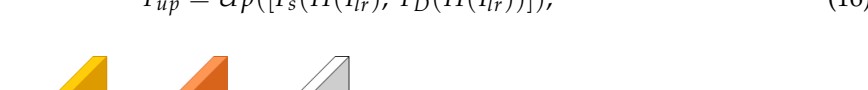

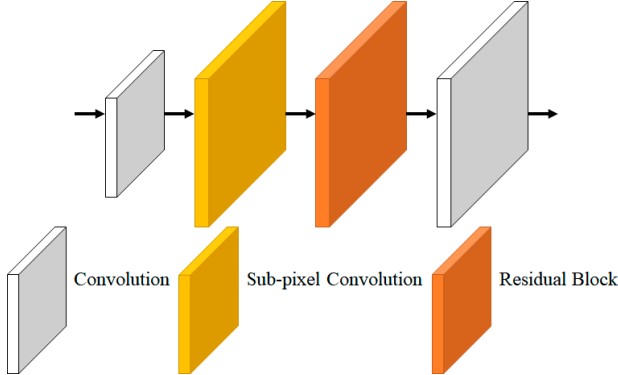

**Figure 12.** Upsampling.

Equation (16) is an equation for extending the feature map, $F_{up}$ is an extended feature map, UP is a sub-pixel convolution, $F_s$ is a three-depth feature, $F_D$ is a 10-depth feature, $I_{lr}$ is input LR image, and $H$ is a convolution. The traditional upsampling method using deconvolution can suffer from the checkerboard artifact problem, which has been addressed by using small strides and large kernel sizes. However, this approach requires a significant number of parameters and computations, which can cause performance degradation. The proposed network expands feature maps using sub-pixel convolution, which avoids the need for a large number of parameters and computations. Sub-pixel convolution was used differently depending on the scale factor to minimize the loss generated in the parameter and expansion process and reduce the amount of computation. Two and three times were extended using one sub-pixel convolution, and in the case of four times, they were extended using two times the sub-pixel convolution.

## 4. Experimental Results

For the network learning proposed in this paper, it was trained using the DIV2K dataset consisting of 800 HR images and the $L_1$ loss.

$$L_1\big(x,\, f(x')\big) = \|x - f(x')\|, \tag{17}$$

Equation (17) is the $L_1$ loss, where $x$ is the original image, $x'$ is the bicubic reduced image, and $f(\cdot)$ is the network of the proposed method. The loss function is used to measure the difference between the ground truth and the output of the network. If the value of the loss function decreases, the prediction is accurate.

Adam optimizer [26] was used for learning, and the parameters used for learning consisted of learning rate, $\beta_1$, and $\beta_2$, respectively. $\beta_1$ is the exponential decay rate for the first-order moment estimate, and $\beta_2$ is the exponential decay rate for the second-order moment estimate. The default value is $10e^{-4}$ for learning rate, 0.9 for $\beta_1$, and 0.999 for $\beta_2$. The input image was composed of $48 \times 48$ patches of LR image size. In this paper, bicubic interpolation, VDSR, and EDSR were used based on the Set5, Set14, B100, and Urban100 datasets to compare the proposed method.

Figure 13 shows an image of the DIV2K Validation PSNR results trained 50 epochs by the number of attention blocks in Show Feature Extraction and Deep Feature Extraction. As the number of attention blocks increased, the performance increased, and when the number of attention blocks exceeded a certain number of attention blocks, the PSNR decreased. Through the experiment, the number of attention blocks in the Show Feature Extraction with the highest PSNR was three and the number of attention blocks in the Deep Feature Extraction was 10 applied.

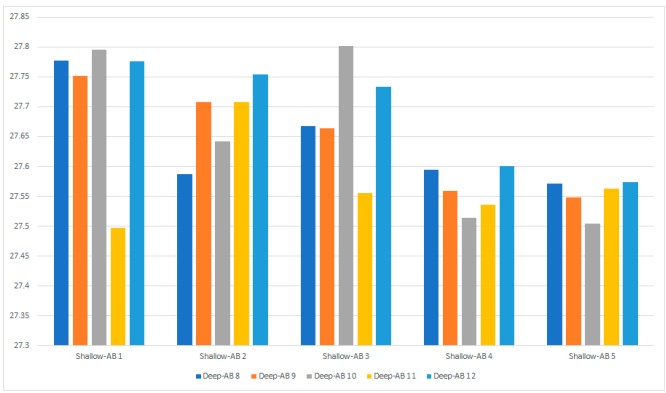

**Figure 13.** Results the number of Attention Blocks.

Figure 14 is a DIV 2K Validation result graph according to the number of CSBlocks constituting the attention block. As the number of CSBlocks increases, the PSNR value increases. However, when the number of CSBlocks exceeds 10, the PSNR value decreases. When a certain level of depth is exceeded, learning difficulties occur, and performance deteriorates even though many parameters are used. For the experiment, 10 CSBlocks with the highest PSNR were applied.

Figure 15 is the four-times result image of img052 of the Urban100 dataset. Peak signal-to-noise ratio (PSNR) is the ratio of noise to the maximum value of the image and is used to measure the image quality loss. The higher the PSNR value, the lower the loss compared to the original video. The structural similarity index measure (SSIM) is a metric developed for the purpose of evaluating human perceptual ability [27]. Human vision is specialized in deriving structural information of images, so structural information has the greatest impact on image quality. Luminance, contrast, and structure are used to measure this, and the larger the value of SSIM, the smaller the loss. Unlike other deep learning algorithms, bicubic interpolation restores the values of 16 neighboring pixels as a two-dimensional formula, so the image is natural, but the result is blurry due to lack of high-frequency region processing. VDSR performed better on building contours than bicubic interpolation using 20 convolutions and residual learning. EDSR shows clear results by using 69 convolutions that are deeper than existing VDSR. The results obtained through proposed CSBlock and attention block methods show improved results in the high-frequency region compared to

existing networks. However, numerically, it shows a value similar to EDSR. However, in the case of EDSR, there is a lack of emphasis on specific high-frequency features because it is a structure that only builds convolutions. However, the method we propose optimizes the network and has the advantage of enabling high-speed learning as well as specific high-frequency regions.

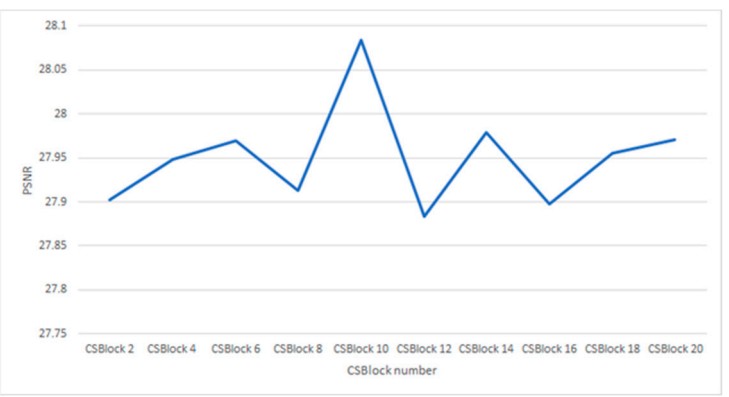

**Figure 14.** Results of the number of CSBlocks.



**Figure 15.** Visual evaluation for a scale factor of $4\times$ on the image "img052" from Urban100: (**a**) Ground Truth (PSNR, SSIM); (**b**) Bicubic (28.24, 0.7311); (**c**) VDSR (32.57, 0.8875); (**d**) EDSR (35.11, 0.9319); (**e**) Proposed **(35.13, 0.9322)**.

Figure 16 is the result of magnifying 8023 of B100 by four times. In the process of restoring the part where the straight pattern is gathered, the existing bicubic interpolation shows that the bird's wing pattern is lost, and the pattern is restored in various directions rather than a straight pattern. In the case of VDSR, it is clearer than bicubic and shows improved results, but all patterns are not restored and the results are restored in the other direction. EDSR shows clear results and direction of patterns with many parameters and deep network structure. The proposed method shows the result that the bird's wing pattern is seamlessly connected.



**Figure 16.** Visual evaluation for a scale factor of $4\times$ on the image "8023" from B100: (**a**) Ground Truth (PSNR, SSIM); (**b**) Bicubic (30.70, 0.7447); (**c**) VDSR (34.28, 0.8699); (**d**) EDSR (35.85, 0.8832); (**e**) Proposed **(35.91, 0.8851)**.

Figure 17 is the result of magnifying PPT3 of Set14 by four times. In bicubic interpolation, the boundary line of the curved part of the letter was restored blurry. VDSR has clearly restored boundary lines, but shows distorted results in some curves. The EDSR method

shows a better restoration result from distortion than the existing VDSR. The proposed method also shows improved results on the contour.

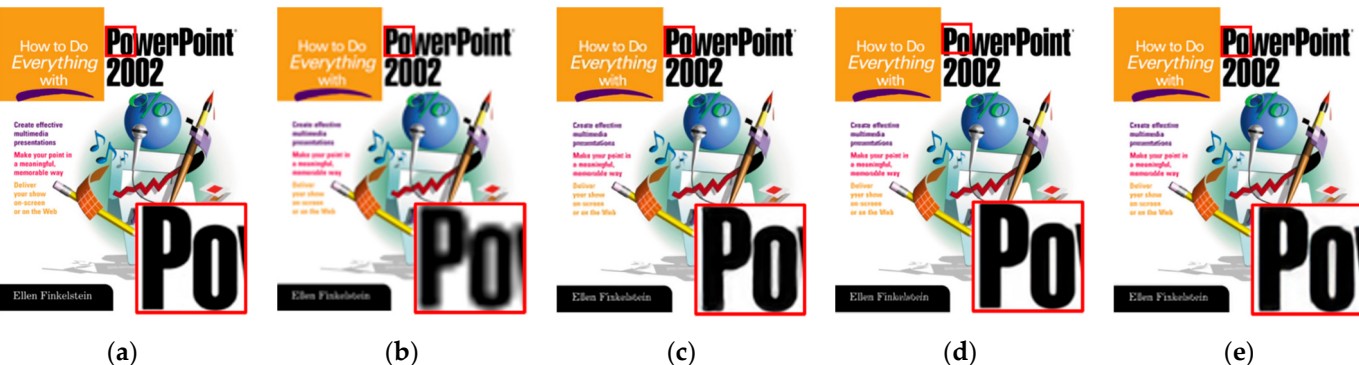

**Figure 17.** Visual evaluation for a scale factor of 4× on the image "PPT3" from Set14: (**a**) Ground Truth (PSNR, SSIM); (**b**) Bicubic (24.31, 0.7889); (**c**) VDSR (29.93, 0.9344); (**d**) EDSR (31.51, 0.9594); (**e**) Proposed **(31.53, 0.9593)**.

Figure 18 is the result of magnifying the COMIC of Set14 by four times. Bicubic interpolation has blurred outlines due to complex and small patterns, and the small patterns have disappeared. VDSR is improved over bicubic interpolation in the small missing patterns of objects, but the outlines are blurry. EDSR contours have been improved and object behavior has been restored. The proposed method has improved contour lines and shows good numerical results.

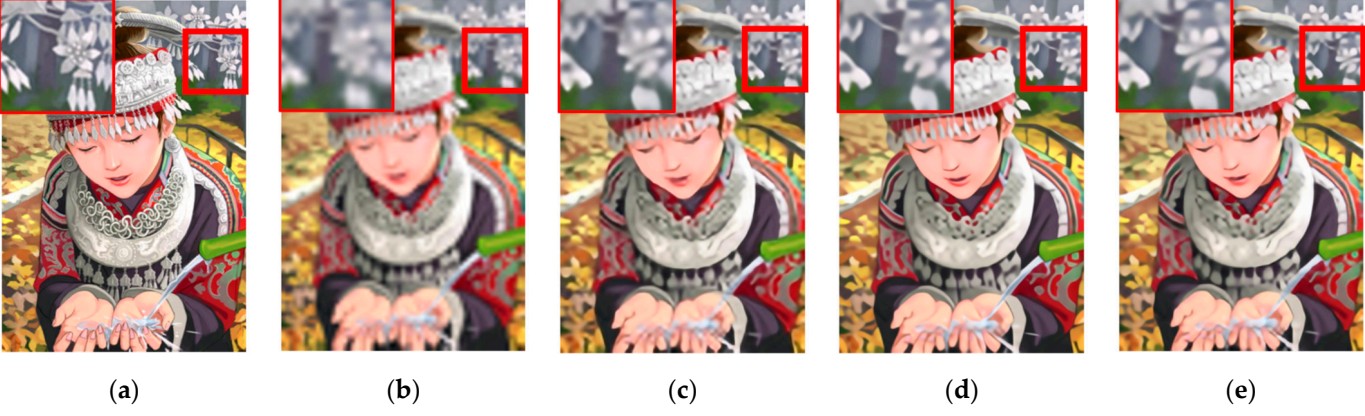

**Figure 18.** Visual evaluation for a scale factor of 4× on the image "COMIC" from Set14: (**a**) Ground Truth (PSNR, SSIM); (**b**) Bicubic (24.05, 0.4994); (**c**) VDSR (27.26, 0.6988); (**d**) EDSR (28.42, 0.7606); (**e**) Proposed **(28.55, 0.7613)**.

Table 1 is a table comparing PNSR and SSIM for each dataset. The first Set5 is a dataset consisting of five images. The image of Set5 is a dataset with a simple pattern and shows higher results in PSNR and SSIM than other datasets in most algorithms. The results obtained through the proposed method show a difference of more than 4%. The second Set14 is a dataset consisting of 14 images of various environments. It is an image dataset similar to Set5 and has lower PNSR and SSIM values than Set5 dataset due to various environments and a more complex dataset than Set5, and showed similar results to Set5. The third Urban100 is a dataset consisting of 100 architectural images. Due to the nature of building images, there are small patterns such as small windows and bricks, and it is important to use high frequencies and estimate loss areas compared to other datasets. The proposed method shows higher performance than the existing EDSR and VDSR methods by utilizing high-frequency components and various features. Finally, B100 is a dataset of 100 objects composed of various shapes such as plants, people, and food in natural

images. Various objects have a lot of irregular shapes rather than regular shapes like the Urban100 dataset, so learning of various patterns has a lot of influence. When using the proposed method, it shows higher performance than the existing network. As a result, our proposed method shows better performance than existing algorithms when there are many high-frequency regions and various patterns exist.

**Table 1.** Average PSNR/SSIM for scale factor ×2, ×3, ×4. Highlighted indicates the best performance.

| Dataset | Scale | Bicubic PNSR/SSIM | VDSR PNSR/SSIM | EDSR PNSR/SSIM | Proposed Method PNSR/SSIM |
|---|---|---|---|---|---|
| Set 5 | ×2 | 33.66/0.9299 | 37.53/0.9587 | 38.11/**0.9601** | **38.15**/0.9589 |
| | ×3 | 30.39/0.8682 | 33.66/0.9213 | **34.65**/0.9282 | 34.48/**0.9291** |
| | ×4 | 28.42/0.8104 | 31.35/0.8838 | 32.46/0.8968 | **32.48**/**0.8988** |
| Set 14 | ×2 | 30.24/0.8688 | 33.03/0.9124 | 33.92/0.9195 | **34.02**/**0.9211** |
| | ×3 | 27.55/0.7742 | 29.77/0.8314 | **30.52**/**0.8462** | 30.43/0.8431 |
| | ×4 | 26.00/0.7027 | 28.01/0.7674 | **28.80**/**0.7876** | 28.77/0.7866 |
| Urban **100** | ×2 | 26.88/0.8403 | 30.76/0.9140 | 32.93/0.9351 | **32.97**/**0.9352** |
| | ×3 | 24.46/0.7349 | 27.14/0.8279 | **28.80**/**0.8653** | 28.71/0.8640 |
| | ×4 | 123.14/0.6577 | 25.18/0.7524 | 26.64/0.8033 | **26.68**/**0.8035** |
| B 100 | ×2 | 29.56/0.8431 | 31.90/0.8960 | 35.03/0.9695 | **35.07**/**0.9701** |
| | ×3 | 27.21/0.7385 | 28.82/0.7976 | **31.26**/**0.9340** | 31.22/0.9337 |
| | ×4 | 23.96/0.6577 | 27.29/0.7251 | 29.25/0.9017 | **29.27**/**0.9020** |

## 5. Conclusions

Existing CNN-based super resolution methods have poor high-frequency performance such as outlines and textures because they do not consider high-frequency regions. In addition, existing networks have a greater effect on performance improvement as the number of connections increases, but there is the problem that learning is difficult. To solve this problem, this paper proposed SISR using an attention mechanism that combines channel attention and spatial attention and a feature extraction process with different depths. Convolution was used to extract features from LR images. It is a CSBlock that combines channel attention and spatial attention to emphasize the high-frequency features that have a lot of influence on image quality from the extracted features, and the attention block is composed of 10 CSBlocks. Skip connection was used to effectively solve the gradient vanishing problem. In order to solve the LR insufficient feature problem, different numbers of attention blocks were configured. In order to utilize the extracted features, concatenation was used to connect the features. Afterwards, the feature map was extended using sub-pixel convolution and an HR image was created through residual block and convolution. Through learning using $48 \times 48$ LR image patches and $L_1$ loss, the noise and quality of the super resolution method were improved, and in particular, it showed very good effects on rectilinear patterns, outlines, and object restoration. With the proposed method, high-frequency region features that greatly affect image quality, such as object outlines and textures, were emphasized and various features were extracted. Finally, the proposed method showed about 11% to 26% improvement over the existing bicubic interpolation and about 1 to 2% improvement over VDSR and EDSR.

**Author Contributions:** Investigation, D.L., S.Y.C. and S.L.; Methodology, D.L.; Software, D.L. and K.J.; Supervision, K.S.; Validation, K.J., S.Y.C., S.L. and K.S.; Writing—original draft, D.L.; Writing—review and editing, K.J., S.Y.C., S.L. and K.S. All authors have read and agreed to the published version of the manuscript.

**Funding:** This work was supported by the Institute of Information & Communications Technology Planning & Evaluation (IITP) grant funded by the Korean government (MSIT) (No. 2020-0-00922, Development of holographic stereogram printing technology based on multi-view imaging); Institute of Information & Communications Technology Planning & Evaluation (IITP) grant funded by the

Korea government (MSIT) (No. 2020-0-01846, Research and development of realistic content device technology); the Ministry of Culture, Sports and Tourism; and the Korea Creative Content Agency (Project Number: R2021040083).

**Institutional Review Board Statement:** Not applicable.

**Informed Consent Statement:** Not applicable.

**Data Availability Statement:** Not applicable.

**Conflicts of Interest:** The authors declare no conflict of interest.

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
