# Peer review of "A Study on the Super Resolution Combining Spatial Attention and Channel Attention"

_applsci, doi:10.3390/app13063408_

Round 1
Reviewer 1 Report
The paper has proposed a deep learning architecture for image super-resolution using attention mechanism to address the shortcomings of conventional CNN-based approaches. For this purpose, attention mechanism for both channel and spatial domains have been proposed. The subject is interesting and the provided experimental results are promising. There are, however, a few things that need to be addressed in the manuscript.
1- Please explain the role of H1 convolution operation in eq. 10. According to the text, it is supposed to calculate the correlation between pixels in the feature map. It would be better to provide an equation for this operation.
2- In section 3.4 (Upsampling) It has been stated that checkerboard pattern arises out of redundant calculations in deconvolution process. In reality, it arises due to the insufficient stride and can be fixed by using a much smaller stride compared to the kernel. Please expand this section to provide more explanation of eq. 16
3- Image resolution using attention mechanism has been considered before as well. It is advisable to cite some references related to those works. E.g.
a. Zhu, H.; Xie, C.; Fei, Y.; Tao, H. Attention Mechanisms in CNN-Based Single Image Super-Resolution: A Brief Review and a New Perspective. Electronics 2021, 10, 1187. https://doi.org/10.3390/electronics10101187
b. S. Fang, S. Meng, Y. Cao, J. Zhang and W. Shi, "Adaptive Channel Attention and Feature Super-Resolution for Remote Sensing Images Spatiotemporal Fusion," 2021 IEEE International Geoscience and Remote Sensing Symposium IGARSS, Brussels, Belgium, 2021, pp. 2572-2575, doi: 10.1109/IGARSS47720.2021.9555093.
Author Response
Thank you for reviewing our paper.
I checked all the reviews and proceeded with the correction.
Response to Reviewer 1 Comments
Point 1: Please explain the role of H1 convolution operation in eq. 10. According to the text, it is supposed to calculate the correlation between pixels in the feature map. It would be better to provide an equation for this operation.
Response 1: The role of H1 convolution is explained in 8p.
-(8p) Convolution is a process of calculating surrounding pixels into a kernel, and can use Spatial information. calculates the relation between pixels in the feature map.
Point 2: In section 3.4 (Upsampling) It has been stated that checkerboard pattern arises out of redundant calculations in deconvolution process. In reality, it arises due to the insufficient stride and can be fixed by using a much smaller stride compared to the kernel. Please expand this section to provide more explanation of eq. 16
Response 2: we explained the disadvantages of deconvolution and the sub-pixel convolution in detail.
-(10p) The traditional upsampling method using deconvolution can suffer from the checkerboard artifact problem, which has been addressed by using small strides and large kernel sizes. However, this approach requires a significant number of parameters and computations, which can cause performance degradation. The proposed network expands feature maps using Sub-pixel Convolution, which avoids the need for a large number of parameters and computations.
Point 3: Image resolution using attention mechanism has been considered before as well. It is advisable to cite some references related to those works.
Response 3: Added reference using Attention Mechanism.
-(2p) Recently, an attention-based super-resolution method using such high-frequency feature emphasis has been proposed[10-12].

Reviewer 2 Report
The paper entitled “A Study on the Super Resolution Combining Spatial Attention and Channel Attention” proposed Single Image Super Resolution using an attention mechanism that emphasizes high-frequency features and a feature extraction process with different depths. The manuscript is well organized and not well written and its contribution is almost enough to publish in Applied Sciences. However, some of the ideas those need to be thoroughly clarified:
1. It is strongly recommended that the text be carefully checked and its problems should be fixed.
2. It is needed to compare the proposed method with the state-of-the-art methods (years 2021 and 2022).
3. The Related Works section is weak and more papers should be reviewed in this section.
4. Explain the “seq2seq” in line 111.
5. In line 159, in the sentence “Fig. 6 is the process of sup-pixel convolution, a is the input feature map, b is the…”, you mentioned to a and b in Fig.6, but there is no a and b in this figure.
6. Some typos in the text:
a. Page 1, Line 40, “.” Should be after references. “After the codebook is generated, the …”, should be “After the codebook generation, the …”.
b. Page 1, Line 43, “.” Should be after references in “neighboring pixel values.[3] The second…”.
c. Page 1, Line 44, “.” Should be after references in “through deep learning-based CNN [4]. For interpolation …”, Check all text for such mistakes.
d. Page 2, Line 82, in the sentence “Fig1 is a ResNet structure consisting of 152 convolution layers …”, “Fig1” should be “Fig. 1”.
e. Page 3, Line 91, correct the sentence “(a)is the existing neural network and (b) is the residual…”
f. And more and more…

Author Response
Thank you for reviewing our paper.
I checked all the reviews and proceeded with the correction.
Response to Reviewer 2 Comments
Point 1: It is strongly recommended that the text be carefully checked and its problems should be fixed.
Response 1: The description and expression of the thesis have been corrected.
Point 2: It is needed to compare the proposed method with the state-of-the-art methods (years 2021 and 2022).
Response 2: There is a difference in the learning method between the proposed method and the recent method, and it is difficult to compare directly.
Point 3: The Related Works section is weak and more papers should be reviewed in this section.
Response 3: The description and expression of the thesis have been corrected.
Point 4: Explain the “seq2seq” in line 111.
Response 4: The roles and ideas of seq2seq were further explained.
-(3p) Seq2Seq is a model used in natural language processing, such as machine translation, that uses an encoder-decoder architecture to transform one sequence into another[16]. However, during the process of compressing the encoder-decoder structure of the seq2seq model into a fixed-size vector, loss of information can occur and the gradient vanishing problem can occur when the input is long. To address these issues, Attention can be used effectively to train the model.
Point 5: In line 159, in the sentence “Fig. 6 is the process of sup-pixel convolution, a is the input feature map, b is the…”, you mentioned to a and b in Fig.6, but there is no a and b in this figure.
Response 5: Added a, b, c of the Fig 6 and explanation
-(5p) Figure 6. Spatial Attention Module: (a) Input Feature Map; (b) the feature map resulting from the convolution; (c) feature map rearranged from the convolution feature map.
Point 6: Some typos in the text:
Response 6: Checked for typos and errors and corrected them.

Reviewer 3 Report
This paper describes the single image Super Resolution using an attention mechanism that emphasizes high-frequency features and a feature extraction process with different depths.
The detailed comments are as follows. Please consider them for further revision.
1. Page 5, Sub-section 2.3, formula (6) is written incorrectly.
2. Page 6, Section 3,"Shallow Feature Extraction" and "Deep Feature Extraction" in Figure 7 can be described in more detail.
3. Page 9, in formula (14), why do we choose to use 3 attention blocks instead of other number of attention blocks in Shallow Feature Extraction. It can be explained and proved by experiment.
4. Page 10, in formula (16), parameter Ilr is not introduced.
Author Response
Thank you for reviewing our paper.
I checked all the reviews and proceeded with the correction.
Response to Reviewer 3 Comments
Point 1: Page 5, Sub-section 2.3, formula (6) is written incorrectly.
Response 1: Fixed wrong formula.
-(5p)
Point 2: Page 6, Section 3,"Shallow Feature Extraction" and "Deep Feature Extraction" in Figure 7 can be described in more detail.
Response 2: Added description of shallow feature extraction and deep feature extraction.
-(5-6p) To enable effective training of deep networks, an Attention Block combining CSBlock and Skip Connection was applied. The Attention Block was used to emphasize features using the structure of Shallow Feature Extraction and Deep Feature Extraction with Attention Block, and feature utilization was performed through connections. In this way, both shal-low features including shape information and deep features with emphasized high-frequency components were extracted from the insufficient information of low-resolution.
Point 3: Page 9, in formula (14), why do we choose to use 3 attention blocks instead of other number of attention blocks in Shallow Feature Extraction. It can be explained and proved by experiment
Response 3: In the experiment, the performance according to the number of Attention Blocks explained the reason for setting three.
-(11p) Fig. 13 shows an image of the DIV2K Validation PSNR results trained 50 epochs by the number of attention blocks in Show Feature Extraction and Deep Feature Extraction. As the number of attention blocks increased, the performance increased, and when the number of attention blocks exceeded a certain number of attention blocks, the PSNR de-creased. Through the experiment, the number of Attention Blocks in the Show Feature Ex-traction with the highest PSNR was 3 and the number of Attention Blocks in the Deep Feature Extraction was 10 applied.
Point 4: Page 10, in formula (16), parameter Ilr is not introduced
Response 4: Added description of Ilr that was missing.
-(10p) Equation 16 is an equation for extending the feature map, is an extended feature map, UP is a Sub-pixel Convolution, is a 3-depth feature, is a 10-depth feature, is input LR image and H is a convolution

Round 2
Reviewer 2 Report
The text still has some grammatical mistakes.